# Differentially Private Uniformly Most Powerful Tests for Binomial Data

**Jordan Awan**
Department of Statistics
Penn State University
University Park, PA 16802
awan@psu.edu

**Aleksandra Slavković**
Department of Statistics
Penn State University
University Park, PA 16802
sesa@psu.edu

## Abstract

We derive uniformly most powerful (UMP) tests for simple and one-sided hypotheses for a population proportion within the framework of Differential Privacy (DP), optimizing finite sample performance. We show that in general, DP hypothesis tests can be written in terms of linear constraints, and for exchangeable data can always be expressed as a function of the empirical distribution. Using this structure, we prove a 'Neyman-Pearson lemma' for binomial data under DP, where the DP-UMP only depends on the sample sum. Our tests can also be stated as a post-processing of a random variable, whose distribution we coin "Truncated-Uniform-Laplace" (Tulap), a generalization of the Staircase and discrete Laplace distributions. Furthermore, we obtain exact $p$-values, which are easily computed in terms of the Tulap random variable. We show that our results also apply to distribution-free hypothesis tests for continuous data. Our simulation results demonstrate that our tests have exact type I error, and are more powerful than current techniques.

## 1 Introduction

Differential Privacy (DP), introduced by DMNS06, offers a rigorous measure of disclosure risk. To satisfy DP, a procedure cannot be a deterministic function of the sensitive data, but must incorporate additional randomness, beyond sampling. Subject to the DP constraint, it is natural to search for a procedure which maximizes the utility of the output. Many works address the goal of minimizing the distance between the output of the randomized DP procedure and standard non-private algorithms, but few attempt to infer properties about the underlying population (for some notable exceptions, see related work), which is typically the goal in statistics and scientific research. In this paper, we study the setting where each individual contributes a sensitive binary value, and we wish to infer the population proportion via hypothesis tests, subject to DP. In particular, we derive *uniformly most powerful* (UMP) tests for simple and one-sided hypotheses, optimizing finite sample performance.

UMP tests are fundamental to classical statistics, being closely linked to sufficiency, likelihood inference, and confidence sets. However, finding UMP tests can be hard and in many cases they do not even exist (see Sch96, Section 4.4). Our results are the first to achieve UMP tests under $(\epsilon, \delta)-$DP, and are among the first steps towards a general theory of optimal inference under DP.

**Related work** Vu and Slavković [VS09] are among the first to perform hypothesis tests under DP. They develop private tests for population proportions as well as for independence in $2 \times 2$ contingency tables. In both settings, they fix the noise adding distribution, and use approximate sampling distributions to perform these DP tests. A similar approach is used by Sol14 to develop tests for normally distributed data. The work of VS09 is extended by WLK15 and GLRV16, developing additional tests for multinomial data. To implement their tests, WLK15 develop asymptotic sampling distributions, verifying via simulations that the type I errors are reliable. On the other hand, GLRV16

use simulations to compute an empirical type I error. Uhler et al. [USF13] develop DP chi-squared tests and $p$-values for GWAS data, and derive the exact sampling distribution of their noisy statistic. Working under "Local Differential Privacy," a stronger notion of privacy than DP, GR18 develop multinomial tests based on asymptotic distributions. Given a DP output, She17 and BRC17 develop significance tests for regression coefficients.

Outside the hypothesis testing setting, there is some work on optimal population inference under DP. Duchi et al. [DJW18] give general techniques to derive minimax rates under local DP, and in particular give minimax optimal point estimates for the mean, median, generalized linear models, and nonparametric density estimation. Karwa and Vadhan [KV17] develop nearly optimal confidence intervals for normally distributed data with finite sample guarantees, which could potentially be inverted to give UMP-unbiased tests.

Related work on developing optimal DP mechanisms for general loss functions such as GV16a and GRS09, give mechanisms that optimize symmetric convex loss functions, centered at a real statistic. Similarly, AS18 derive optimal mechanisms among the class of $K$-Norm Mechanisms.

**Our contributions** The previous literature on DP hypothesis testing has a few characteristics in common: 1) nearly all of these proposed methods first add noise to the data, and perform their test as a post-processing procedure, 2) all of the hypothesis tests use either asymptotic distributions or simulations to derive approximate decision rules, and 3) while each procedure is derived intuitively based on classical theory, none show that they are optimal among all possible DP algorithms.

In contrast, in this paper we search over all DP hypothesis tests at level $\alpha$, deriving the *uniformly most powerful* (UMP) test for a population proportion. In Section 3, we show that arbitrary DP hypothesis tests, which report 'Reject' or 'Fail to Reject', can be written in terms of linear inequalities. In Theorem 3.2, we show that for exchangeable data, DP tests need only depend on the empirical distribution. We use this structure to find closed-form DP-UMP tests for simple hypotheses in Theorems 4.5 and 5.2, and extend these results to obtain one-sided DP-UMP tests in Corollary 5.3. These tests are closely tied to our proposed "*Truncated-Uniform-Laplace*" (*Tulap*) distribution, which extends both the discrete Laplace distribution (studied in GRS09), and the Staircase distribution of GV16a to the setting of $(\epsilon, \delta)$-DP. We prove that the Tulap distribution satisfies $(\epsilon, \delta)$-DP in Theorem 6.1. While the tests developed in the previous sections only resulted in the output 'Reject' or 'Fail to Reject', in Section 6, we show that our DP-UMP tests can be stated as a post-processing of a Tulap random variable. From this formulation, we obtain exact $p$-values via Theorem 6.2 and Algorithm 1 which agree with our DP-UMP tests. In Section 7, we show that our results apply to distribution-free hypothesis tests of continuous data. In Section 8, we verify through simulations that our UMP tests have exact type I error, and are more powerful than current techniques.

## 2 Background and notation

We use capital letters to denote random variables and lowercase letters for particular values. For a random variable $X$, we denote $F_X$ as its cumulative distribution function (cdf), $f_X$ as either its probability density function (pdf) or probability mass function (pmf), depending on the context.

For any set $\mathscr{X}$, the $n$-fold cartesian product of $\mathscr{X}$ is $\mathscr{X}^n = \{(x_1, x_2, \ldots, x_n) \mid x_i \in \mathscr{X}\}$. We denote elements of $\mathscr{X}^n$ with an underscore to emphasize that they are vectors. The *Hamming distance* metric on $\mathscr{X}^n$ is $H : \mathscr{X}^n \times \mathscr{X}^n \to \mathbb{Z}^{\geq 0}$, defined by $H(\underline{x}, \underline{x}') = \#\{i \mid x_i \neq x_i'\}$.

Differential Privacy, introduced by DMNS06, provides a formal measure of disclosure risk. The notion of DP that we give in Definition 2.1 more closely resembles the formulation in WZ10, which uses the language of distributions rather than random mechanisms. It is important to emphasize that the notion of Differential Privacy in Definition 2.1 does not involve any distribution model on $\mathscr{X}^n$.

**Definition 2.1** (Differential Privacy: DMNS06, WZ10). Let $\epsilon > 0$, $\delta \geq 0$, and $n \in \{1, 2, \ldots\}$ be given. Let $\mathscr{X}$ be any set, and $(\mathscr{Y}, \mathscr{F})$ be a measurable space. Let $\mathscr{P} = \{P_{\underline{x}} \mid \underline{x} \in \mathscr{X}^n\}$ be a set of probability measures on $(\mathscr{Y}, \mathscr{F})$. We say that $\mathscr{P}$ satisfies $(\epsilon, \delta)$-*Differential Privacy* ($(\epsilon, \delta)$ - DP) if for all $B \in \mathscr{F}$ and all $\underline{x}, \underline{x}' \in \mathscr{X}^n$ such that $H(\underline{x}, \underline{x}') = 1$, we have $P_{\underline{x}}(B) \leq e^\epsilon P_{\underline{x}'}(B) + \delta$.

In Definition 2.1, we interpret $\underline{x} \in \mathscr{X}^n$ as the database we collect, where $\mathscr{X}$ is the set of possible values that one individual can contribute, and $Y \sim P_{\underline{x}}$ as the statistical result we report to the public. With this interpretation, if a set of distributions satisfies $(\epsilon, \delta)$-DP for small values of $\epsilon$ and $\delta$, then if one person's data is changed in the database, the distribution of $Y$ does not change much. Ideally $\epsilon$ is

a small value less than 1, and $\delta \ll \frac{1}{n}$ allows us to disregard events which have small probability. We refer to $(\epsilon, 0)$-DP as pure DP, and $(\epsilon, \delta)$-DP as approximate DP.

The focus of this paper is to find uniformly most powerful (UMP) hypothesis tests, subject to DP. As the output of a DP method is necessarily a random variable, we work with randomized hypothesis tests, which we review in Definition 2.2. Our notation follows that of Sch96, Chapter 4.

**Definition 2.2** (Hypothesis Test). Let $(X_1, \ldots, X_n) \in \mathscr{X}^n$ be distributed $X_i \overset{\text{iid}}{\sim} f_\theta$, where $\theta \in \Theta$. Let $\Theta_0, \Theta_1$ be a partition of $\Theta$. A *(randomized) test* of $H_0 : \theta \in \Theta_0$ versus $H_1 : \theta \in \Theta_1$ is a measurable function $\phi : \mathscr{X}^n \to [0, 1]$. We say a test $\phi$ is at *level* $\alpha$ if $\sup_{\theta \in \Theta_0} \mathbb{E}_{f_\theta} \phi \leq \alpha$. The *power* of $\phi$ at $\theta$ is denoted $\beta_\phi(\theta) = \mathbb{E}_{f_\theta} \phi$.

Let $\Phi$ be a set of tests. We say that $\phi^* \in \Phi$ is the *uniformly most powerful level* $\alpha$ (UMP-$\alpha$) test among $\Phi$ for $H_0 : \theta \in \Theta_0$ versus $H_1 : \theta \in \Theta_1$ if 1) $\sup_{\theta \in \Theta_0} \beta_{\phi^*}(\theta) \leq \alpha$ and 2) for any $\phi \in \Phi$ such that $\sup_{\theta \in \Theta_0} \beta_\phi(\theta) \leq \alpha$ we have $\beta_{\phi^*}(\theta) \geq \beta_\phi(\theta)$, for all $\theta \in \Theta_1$.

In Definition 2.2, $\phi(x)$ is the probability of rejecting the null hypothesis, given that we observe $x \in \mathscr{X}^n$. That is, the output of a test is either 'Reject', or 'Fail to Reject' with respective probabilities $\phi(x)$, and $1 - \phi(x)$. While the condition of $(\epsilon, \delta)$-DP does not involve the randomness of $X$, for hypothesis testing, the level, and power of a test depend on the model for $X$. In Section 3, we study the set of hypothesis tests which satisfy $(\epsilon, \delta)$-DP.

# 3 Problem setup and exchangeability condition

We begin this section by considering arbitrary hypothesis testing problems under DP. Let $\phi : \mathscr{X}^n \to [0, 1]$ be any test. Since the only possible outputs of the mechanism are 'Reject' or 'Fail to Reject' with probabilities $\phi(\underline{x})$ and $1 - \phi(\underline{x})$, the test $\phi$ satisfies $(\epsilon, \delta)$-DP if and only if for all $\underline{x}, \underline{x}' \in \mathscr{X}^n$ such that $H(\underline{x}, \underline{x}') = 1$,

$$\phi(\underline{x}) \leq e^\epsilon \phi(\underline{x}') + \delta \quad \text{and} \quad (1 - \phi(\underline{x})) \leq e^\epsilon (1 - \phi(\underline{x}')) + \delta. \tag{1}$$

**Remark 3.1.** For any simple hypothesis test, where $\Phi_0$ and $\Phi_1$ are both singleton sets, the DP-UMP test $\phi^*$ is the solution to a linear program. If $\mathscr{X}$ is finite, this observation allows one to explore the structure of DP-UMP tests through numerical linear program solvers.

Given the random vector $\underline{X} \in \mathscr{X}^n$, initially it may seem that we need to consider all $\phi$, which are arbitrary functions of $\underline{X}$. However, assuming that $\underline{X}$ is exchangeable, Theorem 3.2 below says that for any DP hypothesis tests, we need only consider tests which are functions of the empirical distribution of $\underline{X}$. In other words, $\phi$ need not consider the order of the entries in $\underline{X}$. This result is reminiscent of De Finetti's Theorem (see Sch96, Theorem 1.48) in classical statistics.

**Theorem 3.2.** *Let $\Theta$ be a set and $\{\mu_\theta\}_{\theta \in \Theta}$ be a set of exchangeable distributions on $\mathscr{X}^n$. Let $\phi : \mathscr{X}^n \to [0, 1]$ be a test satisfying* (1). *Then there exists $\phi' : \mathscr{X}^n \to [0, 1]$ satisfying* (1) *which only depends on the empirical distribution of $X$, such that $\int \phi'(\underline{x}) \, d\mu_\theta = \int \phi(\underline{x}) \, d\mu_\theta$, for all $\theta \in \Theta$.*

*Proof.* Define $\phi'$ by $\phi'(\underline{x}) = \frac{1}{n!} \sum_{\pi \in \sigma(n)} \phi(\pi(\underline{x}))$, where $\sigma(n)$ is the symmetric group on $n$ letters. For any $\pi \in \sigma(n)$, $\phi(\pi(\underline{x}))$ satisfies $(\epsilon, \delta)$-DP. By exchangeability, $\int \phi(\pi(\underline{x})) \, d\mu_\theta = \int \phi(\underline{x}) \, d\mu_\theta$. Since condition 1 is closed under convex combinations, and integrals are linear, the result follows. $\square$

We now state the particular problem which is the focus for the remainder of the paper, where each individual contributes a sensitive binary value to the database. Let $\underline{X} \in \{0, 1\}^n$ be a random vector, where $X_i$ is the sensitive data of individual $i$. We model $\underline{X}$ as $X_i \overset{\text{iid}}{\sim} \text{Bern}(\theta)$, where $\theta$ is unknown. Then the statistic $X = \sum_{i=1}^n X_i \sim \text{Binom}(n, \theta)$ encodes the empirical distribution of $\underline{X}$. By Theorem 3.2, we can restrict our attention to tests which are functions of $X$. Such tests $\phi : \{0, 1, \ldots, n\} \to [0, 1]$ satisfy $(\epsilon, \delta)$ -DP if and only if for all $x \in \{1, 2, \ldots, n\}$,

$$\phi(x) \leq e^\epsilon \phi(x - 1) + \delta \tag{2}$$
$$\phi(x - 1) \leq e^\epsilon \phi(x) + \delta \tag{3}$$

$$(1 - \phi(x)) \le e^\epsilon (1 - \phi(x - 1)) + \delta \tag{4}$$
$$(1 - \phi(x - 1)) \le e^\epsilon (1 - \phi(x)) + \delta. \tag{5}$$

We denote the set of all tests which satisfy (2)-(5) as $\mathscr{D}_{\epsilon,\delta}^n = \{\phi : \phi \text{ satisfies (2)-(5)}\}$.

**Remark 3.3.** For arbitrary DP hypothesis testing problems, the number of constraints generated by (1) could be very large, even infinite, but for our problem we only have $4n$ constraints.

## 4 Simple DP-UMP tests when $\delta = 0$

In this section, we derive the DP-UMP test when $\delta = 0$ for simple hypotheses. In particular, given $n, \epsilon > 0, \alpha > 0, \theta_0 < \theta_1$, and $X \sim \text{Binom}(n, \theta)$, we find the UMP test at level $\alpha$ among $\mathscr{D}_{\epsilon,0}^n$ for testing $H_0 : \theta = \theta_0$ versus $H_1 : \theta = \theta_1$.

Before developing these tests, we introduce the *Truncated-Uniform-Laplace* (Tulap) distribution, defined in Definition 4.1, which is central to all of our main results. To motivate this distribution, recall that GV16a show for general loss functions, adding discrete Laplace noise $L \sim \text{DLap}(e^{-\epsilon})$ to $X$ is optimal under $(\epsilon, 0)$-DP. For this reason, it is natural to consider a test which post-processes $X + L$. However, we know by classical UMP theory that since $X + L$ is discrete, a randomized test is required. Instead of using a randomized test, by adding uniform noise $U \sim \text{Unif}(-1/2, 1/2)$ to $X + L$, we obtain a continuous sampling distribution, from which a deterministic test is available. We call the distribution of $(X + L + U) \mid X$ as $\text{Tulap}(X, b, 0)$. The distribution $\text{Tulap}(X, b, q)$ is obtained by truncating within the central $(1 - q)^{th}$-quantiles of $\text{Tulap}(X, b, 0)$.

In Definition 4.1, we use the *nearest integer function* $[\cdot] : \mathbb{R} \to \mathbb{Z}$. For any real number $t \in \mathbb{R}$, $[t]$ is defined to be the integer nearest to $t$. If there are two distinct integers which are nearest to $t$, we take $[t]$ to be the even one. Note that, $[-t] = -[t]$ for all $t \in \mathbb{R}$.

**Definition 4.1** (Truncated-Uniform-Laplace (Tulap))**.** Let $N$ and $N_0$ be real-valued random variables. Let $m \in \mathbb{R}$, $b \in (0, 1)$ and $q \in [0, 1)$. We say that $N_0 \sim \text{Tulap}(m, b, 0)$ and $N \sim \text{Tulap}(m, b, q)$ if $N_0$ and $N$ have the following cdfs:

$$F_{N_0}(x) = \begin{cases} \frac{b^{-[x-m]}}{1+b} \left( b + (x - m - [x - m] + \frac{1}{2})(1 - b) \right) & \text{if } x \le [m] \\ 1 - \frac{b^{[x-m]}}{1+b} \left( b + ([x - m] - (x - m) + \frac{1}{2})(1 - b) \right) & \text{if } x > [m], \end{cases}$$

$$F_N(x) = \begin{cases} 0 & \text{if } F_{N_0} < q/2 \\ \frac{F_{N_0}(x) - \frac{q}{2}}{1-q} & \text{if } \frac{q}{2} \le F_{N_0}(x) \le 1 - \frac{q}{2} \\ 1 & \text{if } F_{N_0} > 1 - \frac{q}{2}. \end{cases}$$

Note that a Tulap random variable $\text{Tulap}(m, b, q)$ is continuous and symmetric about $m$.

**Remark 4.2.** The Tulap distribution extends the staircase and discrete Laplace distributions as follows: $\text{Tulap}(0, b, 0) \stackrel{d}{=} \text{Staircase}(b, 1/2)$ and $[\text{Tulap}(0, b, 0)] \stackrel{d}{=} \text{DLap}(b)$, where $\text{Staircase}(b, \gamma)$ is the distribution in GV16a. GV16a show that for a real valued statistic $T$ and convex symmetric loss functions centered at $T$, the optimal noise distribution for $\epsilon$-DP is $\text{Staircase}(b, \gamma)$ for $b = e^{-\epsilon}$ and some $\gamma \in (0, 1)$. If the statistic is a count, then GRS09 show that $\text{DLap}(b)$ is optimal. Our results agree with these works when $\delta = 0$, and extend them to the case of arbitrary $\delta$.

Now that we have defined the Tulap distribution, we are ready to develop the UMP test among $\mathscr{D}_{\epsilon,0}^n$ for the simple hypotheses $H_0 : \theta = \theta_0$ versus $H_1 : \theta = \theta_1$. In classical statistics, the UMP for this test is given by the *Neyman-Pearson lemma*, however in the DP framework, our test must satisfy (2)-(5). Within these constraints, we follow the logic behind the Neyman-Pearson lemma as follows. Let $\phi \in \mathscr{D}_{\epsilon,0}^n$. Thinking of $\phi(x)$ defined recursively, equations (2)-(5) give upper and lower bounds for $\phi(x)$ in terms of $\phi(x - 1)$. Since $\theta_1 > \theta_0$, and binomial distributions have a monotone likelihood ratio (MLR) in $X$, larger values of $X$ give more evidence for $\theta_1$ over $\theta_0$. Thus, $\phi(x)$ should be increasing in $x$ as much as possible, subject to (2)-(5). Lemma 4.3 shows that taking $\phi(x)$ to be such a function is equivalent to having $\phi(x)$ be the cdf of a Tulap random variable.

**Lemma 4.3.** *Let $\epsilon > 0$ be given. Let $\phi : \{0, 1, 2, \ldots, n\} \to (0, 1)$. The following are equivalent:*

*1) There exists $m \in (0, 1)$ such that $\phi(0) = m$ and $\phi(x) = \min\{e^\epsilon \phi(x-1), 1 - e^{-\epsilon}(1 - \phi(x-1))\}$ for $x = 1, \ldots, n$.*

2) *There exists* $m \in (0,1)$ *such that* $\phi(0) = m$ *and for* $x = 1, \ldots, n$,

$$\phi(x) = \begin{cases} e^\epsilon \phi(x-1) & \text{if } \phi(x-1) \le \frac{1}{1+e^\epsilon} \\ 1 - e^{-\epsilon}(1 - \phi(x-1)) & \text{if } \phi(x-1) > \frac{1}{1+e^\epsilon}. \end{cases}$$

3) *There exists* $m \in \mathbb{R}$ *such that* $\phi(x) = F_{N_0}(x-m)$ *for* $x = 0, 1, 2, \ldots, n$, *where* $N_0 \sim$ Tulap$(0, b = e^{-\epsilon}, 0)$.

*Proof Sketch.* First show that 1) and 2) are equivalent by checking which constraint is active. Then verify that $F_{N_0}(x-m)$ satisfies the recurrence of 2). This can be done using the properties of the Tulap cdf, stated in Lemma 10.2, found in the Supplementary Material. □

While the form of 1) in Lemma 4.3 is intuitive, the connection to the Tulap cdf in 3) allows for a usable closed-form of the test. This connection with the Tulap distribution is crucial for the development in Section 6, which shows that the test in Lemma 4.3 can be achieved by post-processing $X + N$, where $N$ is distributed as Tulap.

It remains to show that the tests in Lemma 4.3 are in fact UMP among $\mathscr{D}_{\epsilon,0}^n$. The main tool used to prove this is Lemma 4.4, which is a standard result in the classical hypothesis testing theory.

**Lemma 4.4.** *Let* $(\mathscr{X}, \mathscr{F}, \mu)$ *be a measure space and let* $f$ *and* $g$ *be two densities on* $\mathscr{X}$ *with respect to* $\mu$. *Suppose that* $\phi_1, \phi_2 : \mathscr{X} \to [0,1]$ *are such that* $\int \phi_1 f \, d\mu \ge \int \phi_2 f \, d\mu$, *and there exists* $k \ge 0$ *such that* $\phi_1 \ge \phi_2$ *when* $g \ge kf$ *and* $\phi_1 \le \phi_2$ *when* $g < kf$. *Then* $\int \phi_1 g \, d\mu \ge \int \phi_2 g \, d\mu$.

*Proof.* Note that $(\phi_1 - \phi_2)(g - kf) \ge 0$ for almost all $x \in \mathscr{X}$ (with respect to $\mu$). This implies that $\int (\phi_1 - \phi_2)(g - kf) \, d\mu \ge 0$. Hence, $\int \phi_1 g \, d\mu - \int \phi_2 g \, d\mu \ge k \left( \int \phi_1 f \, d\mu - \int \phi_2 f \, d\mu \right) \ge 0$. □

Next we present our key result, Theorem 4.5, which can be viewed as a 'Neyman-Pearson lemma' for binomial data under $(\epsilon, 0)$-DP. We extend this result in Theorem 5.2 for $(\epsilon, \delta)$-DP.

**Theorem 4.5.** *Let* $\epsilon > 0$, $\alpha \in (0,1)$, $0 \le \theta_0 < \theta_1 \le 1$, *and* $n \ge 1$ *be given. Observe* $X \sim$ Binom$(n, \theta)$, *where* $\theta$ *is unknown. Set the decision rule* $\phi^* : \mathbb{Z} \to [0,1]$ *by* $\phi^*(x) = F_{N_0}(x - m)$, *where* $N_0 \sim$ Tulap$(0, b = e^{-\epsilon}, 0)$ *and* $m$ *is chosen such that* $E_{\theta_0} \phi^*(x) = \alpha$. *Then* $\phi^*$ *is UMP-*$\alpha$ *test of* $H_0 : \theta = \theta_0$ *versus* $H_1 : \theta = \theta_1$ *among* $\mathscr{D}_{\epsilon,0}^n$.

*Proof Sketch.* Let $\phi$ be any other test which satisfies (2)-(5) at level $\alpha$. Then, since $\phi^*$ can be written in the form of 1) in Lemma 4.3, there exists $y \in \mathbb{Z}$ such that $\phi^*(x) \ge \phi(x)$ when $x \ge y$ and $\phi^*(x) \le \phi(x)$ when $x < y$. By MLR of the binomial distribution and Lemma 4.4, we have $\beta_{\phi^*}(\theta_1) \ge \beta_\phi(\theta_1)$. □

While the classical Neyman-Pearson lemma results in an acceptance and rejection region, the DP-UMP always has some probability of rejecting the null, due to the constraints (2)-(5). As $\epsilon \uparrow \infty$, the DP-UMP converges to the non-private UMP.

## 5 Simple and one-sided DP-UMP tests when $\delta \ge 0$

In this section, we extend the results of Section 4 to allow for $\delta \ge 0$. We begin by proposing the form of the DP-UMP test for simple hypotheses. As in Section 4, the DP-UMP test is increasing in $x$ as much as (2)-(5) allow. Lemma 5.1 states that such a test can be written as the cdf of a Tulap random variable, where the parameter $q$ depends on $\epsilon$ and $\delta$. We omit the proof of Theorem 5.2, which mimics the proof of Theorem 4.5.

**Lemma 5.1.** *Let* $\epsilon > 0$ *and* $\delta \ge 0$ *be given and set* $b = e^{-\epsilon}$ *and* $q = \frac{2\delta b}{1 - b + 2\delta b}$. *Let* $\phi : \{0, 1, 2, \ldots, n\} \to [0,1]$. *The following are equivalent:*

1) *There exists* $y \in \{0, 1, 2, \ldots, n\}$ *and* $m \in (0,1)$ *such that*

$$\phi(x) = \begin{cases} 0 & \text{if } x < y \\ m & \text{if } x = y \\ \min\{e^\epsilon \phi(x-1) + \delta, \quad 1 - e^{-\epsilon}(1 - \phi(x-1)) + e^{-\epsilon}\delta, \quad 1\} & \text{if } x > y. \end{cases}$$

2) *There exists $y \in \{0, 1, 2, \ldots, n\}$ and $m \in (0, 1)$ such that*

$$\phi(x) = \begin{cases} 0 & \text{if } x < y \\ m & \text{if } x = y \\ e^\epsilon \phi(x-1) + \delta & \text{if } x > y \text{ and } \phi(x-1) \leq \frac{1-\delta}{1+e^\epsilon} \\ 1 - e^{-\epsilon}(1 - \phi(x-1)) + e^{-\epsilon}\delta & \text{if } x > y \text{ and } \frac{1-\delta}{1+e^\epsilon} \leq \phi(x-1) \leq 1-\delta \\ 1 & \text{if } x > y \text{ and } \phi(x-1) > 1-\delta. \end{cases}$$

3) *There exists $m \in \mathbb{R}$ such that $\phi(x) = F_N(x - m)$ where $N \sim \text{Tulap}(0, b, q)$.*

*Proof Sketch.* The equivalence of 1) and 2) only requires determining which constraints are active. To show the equivalence of 2 and 3, we verify that $F_N(x - m)$ satisfies the recurrence of 2), using the expression of $F_N(x)$ in terms of $F_{N_0}(x)$ given in Definition 4.1, and the results of Lemma 4.3. $\quad\square$

**Theorem 5.2.** *Let $\epsilon > 0$, $\delta \geq 0$, $\alpha \in (0, 1)$, $0 \leq \theta_0 < \theta_1 \leq 1$, and $n \geq 1$ be given. Observe $X \sim \text{Binom}(n, \theta)$, where $\theta$ is unknown. Set $b = e^{-\epsilon}$ and $q = \frac{2\delta b}{1-b+2\delta b}$. Define $\phi^* : \mathbb{Z} \to [0, 1]$ by $\phi^*(x) = F_N(x - m)$ where $N \sim \text{Tulap}(0, b, q)$ and $m$ is chosen such that $E_{\theta_0} \phi^*(x) = \alpha$. Then $\phi^*$ is UMP-$\alpha$ test of $H_0 : \theta = \theta_0$ versus $H_1 : \theta = \theta_1$ among $\mathscr{D}^n_{\epsilon,\delta}$.*

So far we have focused on simple hypothesis tests, but since our test only depends on $\theta_0$, and not on $\theta_1$, our test is in fact the DP-UMP for one-sided tests, as stated in Corollary 5.3. Corollary 5.3 also shows that we can use our tests to build DP-UMP tests for $H_0 : \theta \geq \theta_0$ versus $H_1 : \theta < \theta_0$ as well. Hence, Corollary 5.3 is our most general result so far, containing Theorems 4.5 and 5.2 as special cases.

**Corollary 5.3.** *Let $X \sim \text{Binom}(n, \theta)$. Set $\phi^*(x) = F_N(x - m_1)$ and $\psi^*(x) = 1 - F_N(x - m_2)$, where $N \sim \text{Tulap}\left(0, b = e^{-\epsilon}, q = \frac{2\delta b}{1-b+2\delta b}\right)$ and $m_1, m_2$ are chosen such that $E_{\theta_0} \phi^*(x) = \alpha$ and $E_{\theta_0} \psi^*(x) = \alpha$. Then $\phi^*(x)$ is UMP-$\alpha$ among $\mathscr{D}^n_{\epsilon,\delta}$ for testing $H_0 : \theta \leq \theta_0$ versus $H_1 : \theta > \theta_0$, and $\psi^*(x)$ is UMP-$\alpha$ among $\mathscr{D}^n_{\epsilon,\delta}$ for testing $H_0 : \theta \geq \theta_0$ versus $H_1 : \theta < \theta_0$.*

# 6  Optimal one-sided private p-values

For the DP-UMP tests developed in Sections 4 and 5, the output is simply to 'Reject' or 'Fail to Reject' $H_0$. In scientific research, however, $p$-values are often used to weigh the evidence in favor of the alternative hypothesis over the null. Informally, a $p$-value is the smallest level $\alpha$, for which a test outputs 'Reject'. A more formal definition is given in Definition 10.4, in the Supplementary Material.

In this section, we show that our proposed DP-UMP tests can be achieved by post-processing a Tulap random variable. Using this, we develop a differentially private algorithm for releasing a private $p$-value which agrees with the DP-UMP tests in Sections 4 and 5. While we state our $p$-values for one-sided tests, they also apply to simple tests as a special case.

Since our DP-UMP test from Theorem 5.2 rejects with probability $\phi^*(x) = F_N(x - m)$, given $N \sim F_N$, $\phi^*(x)$ rejects the null if and only if $X + N \geq m$. So, our DP-UMP tests can be stated as a post-processing of $X + N$. Theorem 6.1 states that releasing $X + N$ satisfies $(\epsilon, \delta)$-DP. By the post-processing property of DP (see DR14, Proposition 2.1), once we release $X + N$, any function of $X + N$ also satisfies $(\epsilon, \delta)$-DP. Thus, we can compute our private UMP-$\alpha$ tests as a function of $X + N$ for any $\alpha$. The smallest $\alpha$ for which we reject the null is the $p$-value for that test. In fact Algorithm 1 and Theorem 6.2 give a more elegant method of computing this $p$-value.

**Theorem 6.1.** *Let $\mathscr{X}$ be any set, and $T : \mathscr{X}^n \to \mathbb{Z}$, with $\Delta(T) = \sup |T(\underline{x}) - T(\underline{x}')| = 1$, where the supremum is over the set $\{(\underline{x}, \underline{x}') \in \mathscr{X}^n \times \mathscr{X}^n \mid H(\underline{x}, \underline{x}') = 1\}$. Then the set of distributions $\left\{ \text{Tulap}\left(T(\underline{x}), b = e^{-\epsilon}, \frac{2\delta b}{1-b+2\delta b}\right) \Big| \underline{x} \in \mathscr{X}^n \right\}$ satisfies $(\epsilon, \delta)$-DP.*

*Proof Sketch.* Since Tulap random variables are continuous and have MLR in $T(\underline{x})$, by Lemma 10.3 in the Supplementary Material, it suffices to show that for all $t \in \mathbb{R}$, the cdf of a Tulap random variable $F_N(t - T(\underline{x}))$ satisfies (1), with $\phi(\underline{x})$ replaced with $F_N(t - T(\underline{x}))$. This already established in Lemma 5.1, by the equivalence of 1) and 3). $\quad\square$

**Theorem 6.2.** *Let $\epsilon > 0$, $\delta \geq 0$, $X \sim \text{Binom}(n, \theta)$ where $\theta$ is unknown, and $Z|X \sim \text{Tulap}(X, b = e^{-\epsilon}, q = \frac{2\delta b}{1-b+2\delta b})$. Then*

1) *$p(\theta_0, Z) := P(X + N \geq Z \mid Z)$ is a p-value for $H_0 : \theta \leq \theta_0$ versus $H_1 : \theta > \theta_0$, where the probability is over $X \sim \text{Binom}(n, \theta_0)$ and $N \sim \text{Tulap}(0, b, q)$.*

2) *Let $0 < \alpha < 1$ be given. The test $\phi^*(x) = P_{Z \sim \text{Tulap}(x,b,q)}(p(\theta_0, Z) \leq \alpha \mid X)$ is UMP-$\alpha$ for $H_0 : \theta \leq \theta_0$ versus $H_1 : \theta > \theta_0$ among $\mathscr{D}_{\epsilon,\delta}^n$.*

3) *The output of Algorithm 1 is equal to $p(\theta_0, Z)$.*

It follows from Theorem 6.2 that $p(\theta_0, Z)$ is the stochastically smallest possible $p$-value for the hypothesis test $H_0 : \theta \leq \theta_0$ versus $H_1 : \theta > \theta_0$ under $(\epsilon, \delta)$-DP. Note that $1 - p(\theta_0, Z) = P(X + N \leq Z \mid Z)$ is the $p$-value for $H_0 : \theta \geq \theta_0$ versus $H_1 : \theta < \theta_0$, which agrees with the UMP-$\alpha$ test in Corollary 5.3.

---

**Algorithm 1** UMP one-sided $p$-value for binomial data under $(\epsilon, \delta)$-DP

---

INPUT: $n \in \mathbb{N}$, $\theta_0 \in (0, 1)$, $\epsilon > 0$, $\delta \geq 0$, $Z \sim \text{Tulap}\left(X, b = e^{-\epsilon}, q = \frac{2\delta b}{1-b+2\delta b}\right)$,

1: Set $F_N$ as the cdf of $N \sim \text{Tulap}(0, b, q)$
2: Set $\underline{F} = (F_N(0 - Z), F_N(1 - Z), \ldots, F_N(n - Z))^\top$
3: Set $\underline{B} = (\binom{n}{0}\theta_0^0(1 - \theta_0)^{n-0}, \binom{n}{1}\theta_0^1(1 - \theta_0)^{n-1}, \ldots, \binom{n}{n}\theta_0^n(1 - \theta_0)^{n-n})^\top$
OUTPUT: $\underline{F}^\top \underline{B}$

---

To implement Algorithm 1, we must be able to sample a Tulap random variable, which Algorithm 2 provides. The algorithm is based on the expression of $\text{Tulap}(m, b, 0)$ in terms of geometric and uniform variables, and uses rejection sampling when $q > 0$ (see Bis06, Chapter 11 for an introduction to rejection sampling). A detailed proof that the output of this algorithm follows the correct distribution can be found in Lemma 10.1 in the Supplementary Material.

---

**Algorithm 2** Sample from Tulap distribution: $N \sim \text{Tulap}(m, b, q)$

---

INPUT: $m \in \mathbb{R}$, $b \in (0, 1)$, $q \in [0, 1)$.

1: Draw $G_1, G_2 \overset{\text{iid}}{\sim} \text{Geom}(1 - b)$ and $U \sim \text{Unif}(-1/2, 1/2)$
2: Set $N = G_1 - G_2 + U + m$
3: If $F_{N_0}(N) < q/2$ or $F_{N_0}(N) > 1 - q/2$, where $N_0 \sim \text{Tulap}(m, b, 0)$, go to 1:
OUTPUT: $N$

---

**Remark 6.3.** Since we know that releasing $X + N$, where $N$ is a Tulap random variable, satisfies $(\epsilon, \delta)$-DP, we can compute more than just $p$-values by post-processing $X + N$. We can also compute point estimates for $\theta$, derive posterior distribution of $\theta$ given a prior, and compute confidence intervals for $\theta$ as post-processing of $X + N$. In the full version of this paper, we will study each of these objectives, and connect confidence intervals with the DP-UMP tests derived here.

**Remark 6.4.** One may wonder about the asymptotic properties of the DP-UMP test. It is not hard to show that for any fixed $\epsilon > 0$, $\delta$, and $\theta_0 \in (0, 1)$, our proposed DP-UMP test has asymptotic relative efficiency (ARE) of 1, relative to the non-private UMP test (see vdV00, Section 14.3 for an introduction to ARE). Let $X \sim \text{Binom}(n, \theta_0)$. Define the two test statistics as $T_1 = X$ and $T_2 = X + N$, where $N \sim \text{Tulap}(0, b, q)$. The ARE of the DP-UMP relative to the non-private UMP test is $(C_2/C_1)^2$, where $C_i = \lim_{n \to \infty} \left( \frac{d}{d\theta} \mathbb{E}_\theta T_i \Big|_{\theta=\theta_0} \right) \Big/ \sqrt{n \, \text{Var}_{\theta_0}(T_i)}$, for $i = 1, 2$. We compute $\mathbb{E}_\theta T_i = n\theta$, $\text{Var}_{\theta_0}(T_1) = n\theta_0(1 - \theta_0)$, and $\text{Var}_{\theta_0}(T_2) = n\theta_0(1 - \theta_0) + \text{Var}(N)$. Since $\text{Var}(N)$ is a constant, we have that $C_1 = C_2 = (\theta_0(1 - \theta_0))^{-1/2}$.

## 7 Application to distribution-free inference

In this section, we show how our DP-UMP tests for count data can be used to test certain hypotheses for continuous data. In particular, we give a DP version of the sign and median test allowing one to test the median of either paired or independent samples. For an introduction to the sign and median tests, see Sections 5.4 and 6.4 of GC14. Let $\epsilon > 0$ and $\delta \in [0, 1)$ be given, and let $N \sim \text{Tulap}(0, b, q)$ for $b = e^{-\epsilon}$ and $q = \frac{2\delta b}{1-b-2\delta b}$.

**Sign test:** We observe $n$ iid pairs $(X_i, Y_i)$ for $i = 1, \ldots, n$. Then for all $i = 1, \ldots, n$, $X_i \overset{d}{=} X$ and $Y_i \overset{d}{=} Y$ for some random variables $X$ and $Y$. We assume that for any pair $(X_i, Y_i)$ we can determine if $X_i > Y_i$ or not. For simplicity, we also assume that there are no pairs with $X_i = Y_i$. Denote the unknown probability $\theta = P(X > Y)$. We want to test a hypothesis such as $H_0 : \theta \leq \theta_0$ versus $H_1 : \theta > \theta_0$. The sign test uses the test statistic $T = \#\{X_i > Y_i\}$. Since the sensitivity of $T$ is 1, by Theorem 6.1, $T + N$ satisfies $(\epsilon, \delta)$-DP. Note that the test statistic is distributed as $T \sim \text{Binom}(n, \theta)$. Using Algorithm 1, we obtain a private $p$-value for the sign test as a post-processing of $T + N$.

**Median test:** We observe two independent sets of iid data $\{X_i\}_{i=1}^n$ and $\{Y_i\}_{i=1}^n$, where all $X_i$ and $Y_i$ are distinct values, and we have a total ordering on these values. Then there exists random variables $X$ and $Y$ such that $X_i \overset{d}{=} X$ and $Y_i \overset{d}{=} Y$ for all $i$. We want to test $H_0 : \text{median}(X) \leq \text{median}(Y)$ versus $H_1 : \text{median}(X) > \text{median}(Y)$. The median test uses the test statistic $T = \#\{i \mid \text{rank}(X_i) > n\}$, where $\text{rank}(X_i) = \#\{X_j \leq X_i\} + \#\{Y_j \leq X_i\}$. Since the sensitivity of $T$ is 1, by Theorem 6.1, $T + N$ satisfies $(\epsilon, \delta)$-DP. When $\text{median}(X) = \text{median}(Y)$, $T \sim \text{HyperGeom}(n = n, m = n, k = n)$. Using Algorithm 1, with $\underline{B}$ replaced with the pmf of $\text{HyperGeom}(n = n, m = n, k = n)$, we obtain a private $p$-value for the median test as a post-processing of $T + N$.

# 8 Simulations

In this section, we study both the empirical power and the empirical type I error of our DP-UMP test against the normal approximation proposed by VS09. We define the empirical power to be the proportion of times a test 'Rejects' when the alternative is true, and the empirical type I error as the proportion of times a test 'Rejects' when the null is true. For our simulations, we focus on small samples as the noise introduced by DP methods is most impactful in this setting.

In Figure 1, we plot the empirical power of our UMP test, the Normal Approximation from VS09, and the non-private UMP. For each $n$, we generate 10,000 samples from $\text{Binom}(n, .95)$. We privatize each $X$ by adding $N \sim \text{Tulap}(0, e^{-\epsilon}, 0)$ for the DP-UMP and $L \sim \text{Lap}(1/\epsilon)$ for the Normal Approximation. We compute the UMP $p$-value via Algorithm 1 and the approximate $p$-value for $X + L$, using the cdf of $N\left(X, n/4 + 2/\epsilon^2\right)$. The empirical power is given by $(10000)^{-1}\#\{p\text{-value} < .05\}$. The DP-UMP test indeed gives higher power compared to the Normal Approximation, but the approximation does not lose too much power. Next we see that type I error is another issue.

In Figure 2 we plot the empirical type I error of the DP-UMP and the Normal Approximation tests. We fix $\epsilon = 1$ and $\delta = 0$, and vary $\theta_0$. For each $\theta_0$, we generate 100,000 samples from $\text{Binom}(30, \theta_0)$. For each sample, we compute the DP-UMP and Normal Approximation tests at type I error $\alpha = .05$. We plot the proportion of times we reject the null as well as moving average curves. The DP-UMP, which is provably at type I error $\alpha = .05$ achieves type I error very close to .05, but the Normal Approximation has a higher type I error for small values of $\theta_0$, and a lower type I error for large values of $\theta_0$.

# 9 Discussion and future directions

In this paper, we derived uniformly most powerful simple and one-sided tests for binary data among all DP $\alpha$-level tests. Previously, while various hypothesis tests under DP have been proposed, none have satisfied such an optimality criterion. While our initial DP-UMP tests only output 'Reject' or 'Fail to Reject', we showed that they can be achieved by post-processing a noisy sufficient statistic. This allows us to produce private $p$-values which agree with the DP-UMP tests. Our results can also be applied to obtain $p$-values for distribution-free tests, to test some hypotheses about continuous data under DP.

A simple, yet fundamental observation that underlies our results is that DP tests can be written in terms of linear constraints. This idea alone allows for a new perspective on DP hypothesis testing, which is particularly applicable to other discrete problems, such as multinomial models or difference of population proportions. Stating the problem in this form allows for the consideration of all possible DP tests, and allows the exploration of UMP tests through numerical linear program solvers.

While the focus of this work is on hypothesis testing, these results can also be applied to obtain optimal length confidence intervals for binomial data. In fact, classical statistical theory establishes a

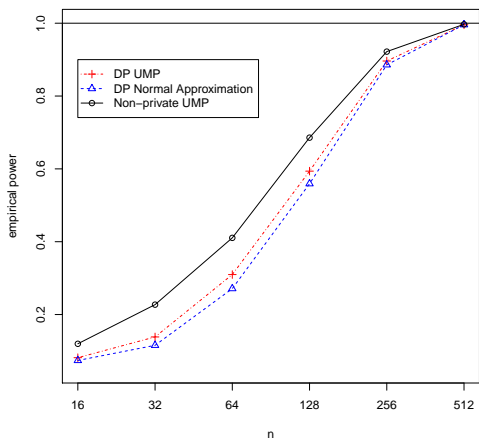

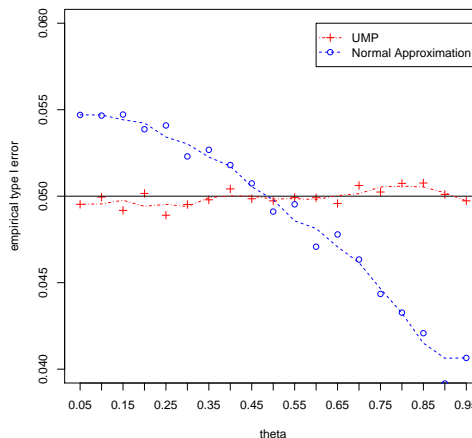

Figure 1: Empirical power for UMP and Normal Approximation tests for $H_0 : \theta \leq .9$ versus $H_1 : \theta \geq .9$. The true value is $\theta = .95$. $\epsilon = 1$ and $\delta = 0$. $n$ varies along the $x$-axis.

Figure 2: Empirical type I error $\alpha$ for UMP and Normal Approximation tests for $H_0 : \theta \leq \theta_0$ versus $H_1 : \theta \geq \theta_0$. $\theta_0$ varies along the $x$-axis. $n = 30$, $\epsilon = 1$, and $\delta = 0$. Target is $\alpha = .05$.

connection between UMP tests and Uniformly Most Accurate (UMA) confidence intervals. Besides confidence intervals, the $p$-value function for the test $H_0 : \theta \geq \theta_0$ versus $H_1 : \theta < \theta_0$ is a cdf which generates a confidence distribution; see XS13 for a review. Since this $p$-value corresponds to the DP-UMP test, this confidence distribution is stochastically more concentrated about the true $\theta$, than any other private confidence distribution. In the full paper, we plan to explore confidence intervals and confidence distributions in detail, establishing connections between our approach here and optimal inference in these settings.

We showed that for exchangeable data, DP tests need only depend on the empirical distribution. For binary data, the empirical distribution is equivalent to the sample sum, which is a complete sufficient statistic for the binomial model. However, in general it is not clear whether optimal DP tests are always a function of complete sufficient statistics as is the case for classical UMP tests. It would be worth investigating whether there is a notion of sufficiency which applies for DP tests.

When $\delta = 0$, our optimal noise adding mechanism, the proposed Tulap distribution, is related to the discrete Laplace distribution, which GRS09 and GV16a also found is optimal for a general class of loss functions. For $\delta > 0$, a truncated discrete Laplace distribution is optimal for our problem. Little previous work has looked into optimal noise adding mechanisms for approximate DP. GV16b studied this problem to some extent, but did not explore truncated Laplace distributions. Steinke [Ste18] proposes that truncated Laplace can be viewed as the canonical distribution for approximate DP in a way that Laplace is canonical for pure DP. Further exploration in the use of truncated Laplace distributions in the approximate DP setting may be of interest.

## Acknowledgements

We would like to thank Vishesh Karwa and Matthew Reimherr for helpful discussions and feedback on previous drafts. We also thank the reviewers for their helpful comments and suggestions, which have contributed to many improvements in the presentation of this work. This work is supported in part by NSF Award No. SES-1534433 to The Pennsylvania State University.

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
