[Supplementary Material]

## 10 Detailed Proofs and Technical Lemmas

*Proof of Theorem 3.2.* Define $\phi'$ by $\phi'(\underline{x}) = \frac{1}{n!}\sum_{\pi \in \sigma(n)} \phi(\pi(\underline{x}))$, where $\sigma(n)$ is the symmetric group on $n$ letters. First note that $\phi(\pi(\underline{x}))$ satisfies (1) for all $\pi \in \sigma(n)$, and that $\int \phi(\pi(\underline{x}))\, d\mu_\theta = \int \phi(\underline{x})\, d\mu_\theta$. Then by exchangeability,

$$\int \phi'(\underline{x})\, d\mu_\theta = \int \frac{1}{n!}\sum_{\pi \in \sigma(n)} \phi(\pi(\underline{x}))\, d\mu_\theta = \frac{1}{n!}\sum_{\pi \in \sigma(n)}\int \phi(\pi(\underline{x}))\, d\mu_\theta$$

$$= \frac{1}{n!}\sum_{\pi \in \sigma(n)}\int \phi(\underline{x})\, d\mu_\theta = \int \phi(\underline{x})\, d\mu_\theta.$$

To see that $\phi'$ satisfies $(\epsilon, \delta)$-DP, we check condition (1):

$$\phi'(\underline{x}) = \frac{1}{n!}\sum_{\pi \in \sigma(n)} \phi(\pi(\underline{x})) \le \frac{1}{n!}\sum_{\pi \in \sigma(n)} (e^\epsilon \phi(\pi(\underline{x}')) + \delta)$$

$$= \frac{1}{n!}\sum_{\pi \in \sigma(n)} e^\epsilon \phi(\pi(\underline{x}')) + \frac{1}{n!}\sum_{\pi \in \sigma(n)} \delta = e^\epsilon \phi'(\underline{x}') + \delta$$

$$(1 - \phi'(\underline{x})) = \left(1 - \frac{1}{n!}\sum_{\pi \in \sigma(n)} \phi(\pi(\underline{x}))\right) = \frac{1}{n!}\sum_{\pi \in \sigma(n)} (1 - \phi(\pi(\underline{x})))$$

$$\le \frac{1}{n!}\sum_{\pi \in \sigma(n)} (e^\epsilon (1 - \phi(\pi(\underline{x}'))) + \delta) = e^\epsilon (1 - \phi'(\pi(\underline{x}'))) + \delta. \qquad \square$$

**Lemma 10.1.** *1) Let $L \sim \mathrm{DLap}(b)$, $U \sim \mathrm{Unif}(-1/2, 1/2)$, $G_1, G_2 \overset{iid}{\sim} \mathrm{Geom}(1-b)$, and $N_0 \sim \mathrm{Tulap}(m, b, 0)$, where the pmf of $L$ is $f_L(x) = \frac{1-b}{1+b}b^{|x|}$ for $x \in \mathbb{Z}$, and the pmf of $G_1$ is $f_{G_1}(x) = (1-p)^x p$ for $x \in \{0, 1, 2, \dots\}$. Then $L + U + m \overset{d}{=} G_1 - G_2 + U + m \overset{d}{=} N_0$.*

*2) Let $N$ be the output of Algorithm 2 with inputs $m, b, q$. Then $N \sim \mathrm{Tulap}(m, b, q)$.*

*3) The random variable $N \sim \mathrm{Tulap}(m, b, q)$ is continuous and symmetric about $m$.*

*Proof of Lemma 10.1.* 1) We know that $L \overset{d}{=} G_1 - G_2$, as shown in IK06. Let $f_U(\cdot)$ denote the pdf of $U$, and $F_U$ denote the cdf of $U$. We will use the property that $f_U(x) = f_U(-x)$ and $F_U(-x) = 1 - F_U(x)$. Then the pdf of $L + U$ is

$$f_{L+U}(x) = f_U(x - [x])\left(\frac{1-b}{1+b}\right)b^{|[x]|} = \begin{cases} f_U(x - [x])\left(\frac{1-b}{1+b}\right)b^{-[x]} & [x] \le 0 \\ f_U(x - [x])\left(\frac{1-b}{1+b}\right)b^{[x]} & [x] > 0. \end{cases}$$

If $[x] \le 0$, then we have

$$F_{L+U}(x) = \int_{-\infty}^{x} f_U(t - [t])\left(\frac{1-b}{1+b}\right)b^{-[t]}\, dt$$

$$= \int_{-\infty}^{[x]-1/2} f_U(t - [t])\left(\frac{1-b}{1+b}\right)b^{-[t]}\, dt + \int_{[x]-1/2}^{x} f_U(t - [x])\left(\frac{1-b}{1+b}\right)b^{-[x]}\, dt$$

$$= \sum_{t=-\infty}^{[x]-1}\left(\frac{1-b}{1+b}\right)b^{-t} + \int_{[x]-1/2}^{x} f_U(t - [x])\left(\frac{1-b}{1+b}\right)b^{-[x]}\, dt$$

$$= \frac{b^{-[x]+1}}{1+b} + F_U(x - [x])\left(\frac{1-b}{1+b}\right)b^{-[x]}$$

$$= \frac{b^{-[x]}}{1+b}(b + F_U(x - [x])(1 - b)).$$

Since, $L + U$ is symmetric about zero, as both $L$ and $U$ are symmetric about zero, for $[x] \geq 0$ we have $F_{L+U}(x) = 1 - F_{L+U}(-x)$. The rest follows by replacing $x$ with $x - m$, and $F_U(x) = x + 1/2$.

2) If $q = 0$, then by part 1), it is clear that the output of Algorithm 2 has the correct distribution. If $q > 0$, then by rejection sampling, we have that $N \sim \mathrm{Tulap}(m, b, q)$. For an introduction to rejection sampling, see Bis06, Chapter 11.

3) This property follows immediately from 1), and that $\mathrm{Tulap}(m, b, q)$ is truncated equally on both sides of $m$. $\square$

**Lemma 10.2.** *Let $N \sim \mathrm{Tulap}(m, b, q)$ and let $t \in \mathbb{Z}$. Then $F_N(t) = \begin{cases} b^{-t} C(m) & t \leq [m] \\ 1 - b^t C(-m) & t > [m], \end{cases}$*

*where $C(m) = (1 + b)^{-1} b^{[m]} (b + ([m] - m + 1/2)(1 - b))$. $C(m)$ is positive, monotone decreasing, and continuous in $m$. Furthermore, $b^{-[m]} C(m) = 1 - b^{[m]} C(-m)$.*

*Proof of Lemma 10.2.* The form of the cdf at integer values is easily verified from Lemma 10.1. It is clear that $C(m)$ is positive. It is also clear that $C(m)$ is continuous and monotone decreasing for all $m \in \mathbb{R} \setminus \{z + 1/2 \mid z \in \mathbb{Z}\}$. So, we will check that $C$ is continuous at $m = z + 1/2$ for $z \in \mathbb{Z}$:

$$\lim_{\epsilon \downarrow 0}(1 + b)C(z + 1/2 + \epsilon) = \lim_{\epsilon \downarrow 0} b^{z+1}(b + (1 - \epsilon)(1 - b)) = b^{z+1}$$

$$\lim_{\epsilon \downarrow 0}(1 + b)C(z + 1/2 - \epsilon) = \lim_{\epsilon \downarrow 0} b^z(b + \epsilon(1 - b)) = b^{z+1}.$$

Since $C$ is continuous on $\mathbb{R}$ and monotone decreasing almost everywhere, it follows that $C$ is monotone decreasing on $\mathbb{R}$ as well.

Call $\alpha(m) = [m] - m + 1/2$, which lies in $[0, 1]$. Note that $\alpha(-m) = -[m] + m + 1/2 = 1 - \alpha(m)$. Then

$$(1 + b)b^{-[m]} C(m) = b + \alpha(m)(1 - b) = b + (1 - \alpha(-m)(1 - b)) = b + (1 - b) - \alpha(-m)(1 - b)$$

$$= (1 + b) - (b + \alpha(-m)(1 - b)) = (1 + b)(1 - b^{[m]} C(-m)). \qquad \square$$

*Proof of Lemma 4.3.* First we show that 1) and 2) are equivalent. Clearly the $m$ is the same for both. We must show that for $p \in (0, 1)$, $e^\epsilon p \leq 1 - e^{-\epsilon}(1 - p)$ whenever $p \leq \frac{1}{1+e^\epsilon}$, and $e^\epsilon p > 1 - e^{-\epsilon}(1 - p)$ when $p > \frac{1}{1+e^\epsilon}$. Setting equal $e^\epsilon p = 1 - e^{-\epsilon}(1 - p)$ we find that $p = \frac{1}{1+e^\epsilon}$. As $p \to 1$, we have that $e^\epsilon p > 1 - e^{-\epsilon}(1 - p)$ and as $p \to 0$, we have $e^\epsilon p < 1 - e^{-\epsilon}(1 - p)$. We conclude that 1) and 2) are equivalent.

Next we show that 2) and 3) are equivalent. First we show that $F_{N_0}(x - m)$ satisfies the recurrence relation in 2). Set $b = e^{-\epsilon}$. First we show that for $t \in \mathbb{Z}$ such that $t \leq [m] - 1$, $F_{N_0}(t - m) \leq \frac{1}{1+e^\epsilon}$ and for $t \geq [m]$, $F_{N_0}(t - m) \geq \frac{1}{1+e^\epsilon}$. Since, $F_{N_0}(\cdot - m)$ is increasing, it suffices to check $t = [m] - 1$ and $t = [m]$:

$$F_{N_0}([m] - 1 - m) = b^{-[m]+1+[m]} \frac{(b + ([m] - m + 1/2)(1 - b))}{1 + b} \leq \frac{b}{1 + b} = \frac{1}{1 + e^\epsilon}$$

$$F_{N_0}([m] - m) = b^{-[m]+[m]} \frac{(b + ([m] - m + 1/2)(1 - b))}{1 + b} \geq \frac{b}{1 + b} = \frac{1}{1 + e^\epsilon},$$

where we use the fact that $0 \leq [m] - m + 1/2 \leq 1$. Now, let $t \in \mathbb{Z}$ and check three cases:

- Let $t < [m]$, then $e^\epsilon F_{N_0}(t - m) = e^\epsilon b^{-t} C(m) = b^{-(t+1)} C(m) = F_{N_0}(t + 1 - m)$.

- Let $t = [m]$. Using Lemma 10.2, $1 - e^{-\epsilon}(1 - F_{N_0}(t - m)) = 1 - b(1 - b^{-[m]} C(m)) = 1 - b + b(1 - b^{[m]} C(-m)) = 1 - b + b - b^{[m+1]} C(-m) = F_{N_0}(t + 1 - m)$.

- Let $t > m$. Then $1 - e^{-\epsilon}(1 - F_{N_0}(t - m)) = 1 - b(b^t C(-m)) = 1 - b^{t+1} C(-m) = F_{N_0}(t + 1 - m)$.

Finally, for any value $c \in (0, 1)$, we can find $m$ such that $F_{N_0}(0 - m) = c$, by the intermediate value theorem. On the other hand, given $m$, set $\phi(0) = F_{N_0}(0 - m)$. $\square$

*Proof of Lemma 4.4.* Note that $(\phi_1 - \phi_2)(g - kf) \geq 0$ for almost all $x \in \mathcal{X}$ (with respect to $\mu$). Then $\int (\phi_1 - \phi_2)(g - kf) \, d\mu \geq 0$. Hence, $\int \phi_1 g \, d\mu - \int \phi_2 g \, d\mu \geq k(\int \phi_1 f \, d\mu - \int \phi_2 f \, d\mu) \geq 0$. $\quad\square$

*Proof of Theorem 4.5.* First note that $\phi^* \in \mathscr{D}_{\epsilon,0}^n$, since by Lemma 4.3, $\phi^*(x) = \min\{e^\epsilon \phi^*(x - 1), 1 - e^{-\epsilon}(1 - \phi^*(x - 1))\}$. So, $\phi^*$ satisfies (2)-(5). Next, since by Lemma 10.2, $F_{N_0}(x - m)$ is a continuous, decreasing function in $m$ with $\lim_{m \uparrow \infty} F_{N_0}(x - m) = 0$ and $\lim_{m \downarrow -\infty} F_{N_0}(x - m) = 1$, we can find $m$ such that $E_{\theta_0} \phi^*(x) = \alpha$ by the Intermediate Value Theorem.

Now that we have argued that $\phi^*$ is a valid test, the rest of the result is an application of Lemma 4.4. It remains to show that the assumptions are satisfied for the lemma to apply. Let $\phi \in \mathscr{D}_{\epsilon,0}^n$ such that $E_{\theta_0} \phi(x) \leq \alpha$.

We claim that either $\phi(x) = \phi^*(x)$ for all $x \in \{0, 1, 2, \ldots, n\}$ or there exists $y$ such that $\phi(y) < \phi^*(y)$. To the contrary, suppose that $\phi^*(x) \leq \phi(x)$ for all $x$ and there exists $z$ such that $\phi^*(z) < \phi(z)$. But this implies that $\phi^*(0) < \phi(0)$ (as we implied by the following paragraphs, by setting $y = 0$). Then $E_{\theta_0} \phi^*(x) < E_{\theta_0} \phi(x) \leq \alpha$ since the pmf of $\mathrm{Binom}(n, \theta)$ is nonzero at $x = 0$, contradicting the fact that $E_{\theta_0} \phi^*(x) = \alpha$. We conclude that there exists $y$ such that $\phi^*(y) > \phi(y)$.

Let $y$ be the smallest point in $\{0, 1, 2, \ldots, n\}$ such that $\phi^*(y) \geq \phi(y)$. We claim that for all $x \geq y$, we have $\phi^*(x) \geq \phi(x)$. We already know that for $y = x$, the claim holds. For induction, suppose the claim holds for some $x \geq y$. By Lemma 4.3, we know that $\phi^*(x + 1) = \min\{e^\epsilon \phi^*(x), 1 - e^{-\epsilon}(1 - \phi^*(x))\}$, and by constraints (2)-(5), we know that $\phi(x + 1) \leq \min\{e^\epsilon \phi(x), 1 - e^{-\epsilon}(1 - \phi(x))\}$.

- Case 1: If $\phi^*(x) \leq \frac{1}{1+e^\epsilon}$, then by Lemma 4.3, $\phi^*(x+1) = e^\epsilon \phi^*(x) \geq e^\epsilon \phi(x) \geq \phi(x+1)$.

- Case 2: If $\phi^*(x) > \frac{1}{1+e^\epsilon}$, then by Lemma 4.3, $\phi^*(x + 1) = 1 - e^{-\epsilon}(1 - \phi^*(x)) \geq 1 - e^{-\epsilon}(1 - \phi(x)) \geq \phi(x+1)$.

We conclude that $\phi^*(x + 1) \geq \phi(x + 1)$. By induction, the claim holds for all $x \in \{y, y + 1, y + 2, \ldots, n\}$. So, we have that $\phi^*(x) \geq \phi(x)$ for $x \in \{y, y + 1, y + 2, \ldots, n\}$ and $\phi^*(x) \leq \phi(x)$ for $x \in \{0, 1, 2, \ldots, y - 1\}$. Since $\mathrm{Binom}(n, \theta)$ has a monotone likelihood ratio in $\theta$, by Lemma 4.4 we have that $E_{\theta_1} \phi^*(x) \geq E_{\theta_1} \phi(x)$. We conclude that $\phi^*$ is UMP-$\alpha$ among $\mathscr{D}_{\epsilon,0}^n$ for the stated hypothesis test. $\quad\square$

*Proof of Lemma 5.1.* We will abbreviate $F(x) := F_{N_0}(x - m)$, where $N_0 \sim \mathrm{Tulap}(0, b = e^{-\epsilon}, 0)$ to simplify notation. First we will show that 1) and 2) are equivalent. It is clear that $y$ and $m$ are the same in both. Next consider $1 - e^{-\epsilon}(1 - p) + e^{-\epsilon}\delta = e^\epsilon p + \delta$, solving for $p$ gives $p = \frac{1-\delta}{1+e^\epsilon}$. Considering as $p \to 0$ and $p \to 1$, we see that $1 - e^{-\epsilon}(1 - p) + e^{-\epsilon}\delta \geq e^\epsilon p + \delta$ when $p \leq \frac{1-\delta}{1+e^\epsilon}$ and $1 - e^{-\epsilon}(1 - p) + e^{-\epsilon}\delta \leq e^\epsilon p + \delta$ when $p \geq \frac{1-\delta}{1+e^\epsilon}$.

Next solving $1 - e^{-\epsilon}(1 - p) + e^{-\epsilon}\delta = 1$ for $p$ gives $p = 1 - \delta$. So, $1 - e^{-\epsilon}(1 - p) + e^{-\epsilon}\delta \leq 1$ when $p \leq 1 - \delta$ and $1 - e^{-\epsilon}(1 - p) + e^{-\epsilon}\delta \geq 1$ when $p \geq 1 - \delta$. Lastly, solving $e^\epsilon p + \delta = 1$ for $p$ gives $p = \frac{1-\delta}{e^\epsilon} \geq \frac{1-\delta}{1+e^\epsilon}$. Combining all of these comparisons, we see that 1) is equivalent to 2).

Before we justify the equivalence of 2) and 3), we argue the following claim. Let $\phi(x)$ be defined as in 3). Then $\phi(x) \leq \frac{1-\delta}{1+e^\epsilon}$ if and only if $F(x) \leq \frac{1}{1+e^\epsilon}$. Suppose that $\phi(x) \leq \frac{1-\delta}{1+e^\epsilon}$. Then $\frac{F(x)-q/2}{1-q} \leq \frac{1-\delta}{1+e^\epsilon}$. Thus,

$$
\begin{aligned}
F(x) &\leq \frac{(1-q)(1-\delta)}{1+e^\epsilon} + \frac{q}{2} \\
&= \frac{1}{1+e^\epsilon}\left((1-q)(1-\delta) + \left(\frac{b+1}{b}\right)\frac{q}{2}\right) \\
&= \frac{1}{1+e^\epsilon}\left(\frac{(1-b)(1-\delta)}{1-b+2\delta b} + \left(\frac{b+1}{b}\right)\frac{\delta b}{1-b+2\delta b}\right) \\
&= \frac{1}{1+e^\epsilon}(1-b+2\delta b)^{-1}((1-b)(1-\delta) + (b+1)\delta) \\
&= \frac{1}{1+e^\epsilon}.
\end{aligned}
$$

We are now ready to show that $\phi(x)$ as described in 3) fits the form of 2).

- Suppose that $0 < \phi(x) < \frac{1-\delta}{1+e^\epsilon}$. By the above, we know that $F(x) \leq \frac{1}{1+e^\epsilon}$. By Lemma 4.3,

$$e^\epsilon \phi(x) + \delta = \frac{e^\epsilon F(x) - \frac{q}{2b}}{1-q} + \delta = \frac{F(x+1) - \frac{q}{2}}{1-q} + \frac{\frac{q}{2} - \frac{q}{2b}}{1-q} + \delta$$

$$= \phi(x+1) + \frac{\delta b}{1-b}\left(1 - \frac{1}{b}\right) + \delta = \phi(x+1).$$

- Suppose that $\frac{1-\delta}{1+e^\epsilon} < \phi(x) \leq 1 - \delta$. Then we have $F(x) > \frac{1}{1+e^\epsilon}$. Then

$$1 - e^{-\epsilon}(1 - \phi(x)) + e^{-\epsilon}\delta = 1 - e^{-\epsilon}\left(1 - \frac{F(x) - q/2}{1-q}\right) + e^{-\epsilon}\delta$$

$$= (1-q)^{-1}\left(1 - q - e^{-\epsilon}(1 - F(x) - q/2)\right) + e^{-\epsilon}\delta$$

$$= (1-q)^{-1}(1 - e^{-\epsilon}(1 - F(x)) + bq/2 - q) + b\delta$$

$$= (1-q)^{-1}(F(x+1) - q/2) + \frac{(b-1)q/2}{1-q} + b\delta$$

$$= \phi(x+1) + \frac{\delta b(b-1)}{1-b} + b\delta$$

$$= \phi(x+1).$$

- Finally, we must show that if $\phi(x) = 1$ then $\phi(x-1) \geq 1 - \delta$. It suffices to show that $F(x) \geq 1 - q/2$ implies that $F(x-1) \geq (1-\delta)(1-q) + q/2 = 1 - (1/b)(q/2)$. We prove the contrapositive. Suppose that $F(x-1) < 1 - (1/b)(q/2)$. Then since $F$ satisfies property (4), we know that

$$F(x) \leq 1 - e^{-\epsilon}(1 - F(x-1)) < 1 - b(1 - (1 - (1/b)(q/2)))$$
$$= 1 - b(1 - 1 + (1/b)(q/2)) = 1 - q/2.$$

We have justified that $\phi(x)$ in 3) satisfies the recurrence relation in 2). Given $\phi'$ of the form in 2), with first non-zero entry at $y$, by Lemma 10.2 and Intermediate Value Theorem, we can find $m \in \mathbb{R}$ such that $\phi(y) = \phi'(y)$. We conclude that 1), 2), and 3) are all equivalent. $\qquad\square$

*Proof of Corollary 5.3.* First we show that $\phi^*$ is UMP-$\alpha$ for $H_0 : \theta \leq \theta_0$ versus $H_1 : \theta > \theta_0$. Since $\phi^*(x)$ is increasing and $\mathrm{Binom}(n, \theta)$ has a monotone likelihood ratio in $\theta$, $E_\theta \phi^* \leq E_{\theta_0} \phi^* = \alpha$ for all $\theta \leq \theta_0$ (property of MLR). By Theorem 4.5, we know that $\phi^*(x)$ is most powerful for any alternative $\theta_1 > \theta_0$ versus the null $\theta_0$. So, $\phi^*$ is UMP-$\alpha$.

Next we show that $\psi^*$ is UMP-$\alpha$ for $H_1 : \theta \geq \theta_0$ versus $H_1 : \theta < \theta_0$. First note that $\sup_{\theta \geq \theta_0} \mathbb{E}_\theta \psi^* = \alpha$. Let $\psi$ be another test with $\sup_{\theta \geq \theta_0} \mathbb{E}_\theta \psi \leq \alpha$. Let $\theta_1 < \theta_0$, we will show that $\mathbb{E}_{\theta_1} \psi^* \geq \mathbb{E}_{\theta_1} \psi$. Define $\widetilde{\psi}^*(x) = \psi^*(n-x) = 1 - F_{N_0}(n-x-m_2) = F_{N_0}(x+m_2-n)$ and $\widetilde{\psi}(x) = \psi(n-x)$. Then using the map $(x, \theta) \mapsto (n-x, 1-\theta)$, we have that $\mathbb{E}_{X \sim (1-\theta_0)} \widetilde{\psi}^*(X) = \mathbb{E}_{X \sim (1-\theta_0)} \psi^*(n-X) = \mathbb{E}_{Y \sim \theta_0} \psi^*(Y) = \alpha$. By a similar argument for $\psi$, we have that both $\widetilde{\psi}^*$ and $\widetilde{\psi}$ are level $\alpha$ for $H_0 : \theta \leq 1 - \theta_0$ versus $H_1 : \theta > 1 - \theta_0$. Since $\mathbb{E}_{(1-\theta_0)} \widetilde{\psi}^* = \alpha$, and $\widetilde{\psi}^*(x) = F_{N_0}(x - m')$, we have that $\widetilde{\psi}^*$ is UMP-$\alpha$ for $H_0 : \theta \leq (1 - \theta_0)$ versus $H_1 : \theta > (1 - \theta_0)$. Then for $\theta_1 < \theta_0$,

$$\mathbb{E}_{X \sim \theta_1} \psi^*(X) = \mathbb{E}_{Y \sim (1-\theta_1)} \widetilde{\psi}^*(Y) \geq \mathbb{E}_{Y \sim (1-\theta_1)} \widetilde{\psi}(Y) = \mathbb{E}_{X \sim \theta_1} \psi(X).$$

We conclude that $\psi^*$ is UMP-$\alpha$ for $H_1 : \theta \geq \theta_0$ versus $H_1 : \theta < \theta_0$. $\qquad\square$

**Lemma 10.3.** *Observe $\underline{x} \in \mathscr{X}^n$. Let $T : \mathscr{X}^n \to \mathbb{R}$, and let $\{\mu_{\underline{x}} \mid \underline{x} \in \mathscr{X}^n\}$ be a set of probability measures on $\mathbb{R}$, dominated by Lebesgue measure. Suppose that $\mu_{\underline{x}}$ is parameterized by $T(\underline{x})$ and $\mu_{\underline{x}}$ has MLR in $T(\underline{x})$. Then $\{\mu_{\underline{x}}\}$ satisfies $(\epsilon, \delta)$-DP if and only if for all $H(\underline{x}_1, \underline{x}_2) = 1$ and all $t \in \mathbb{R}$,*

$$\mu_{\underline{x}_1}((-\infty, t)) \leq e^\epsilon \mu_{\underline{x}_2}((-\infty, t)) + \delta, \tag{6}$$

$$\mu_{\underline{x}_1}((t, \infty)) \leq e^\epsilon \mu_{\underline{x}_2}((t, \infty)) + \delta. \tag{7}$$

*Proof of Lemma 10.3.* Let $\alpha \in [0,1]$ be given. We will only consider $B \subset \mathbb{R}$ (Lebesgue measurable) such that $\mu_{\underline{x}_2}(B) = \alpha$. Then demonstrating $(\epsilon, \delta)$-DP requires $\sup\limits_{\{B | \mu_{\underline{x}_2}(B) = \alpha\}} \mu_{\underline{x}_1}(B) \leq e^\epsilon \alpha + \delta$. We interpret this problem as testing the hypothesis $H_0 : \underline{x} = \underline{x}_2$ versus $H_1 : \underline{x} = \underline{x}_1$, using the rejection region $B$, where $\alpha$ is the type I error, and $\mu_{\underline{x}_1}(B)$ is the power. We know that $\sup\limits_{\{B | \mu_{\underline{x}_2}(B) = \alpha\}} \mu_{\underline{x}_1}(B)$ is achieved by the Neyman-Pearson Lemma. Since $\mu_{\underline{x}}$ has an MLR in $T(\underline{x})$, $\arg\sup_{\{B | \mu_{\underline{x}_2}(B) = \alpha\}} \mu_{\underline{x}_1}(B)$ is either of the form $(-\infty, t)$ or $(t, \infty)$, depending on whether $T(\underline{x}_1)$ is greater or lesser than $T(\underline{x}_2)$. Since $\mu_{\underline{x}_1}$ is dominated by Lebesgue measure for all $\underline{x}_1$, $\mu_{\underline{x}_2}((-\infty, t))$ is continuous in $t$, which allows us to achieve exactly $\alpha$ type I error. $\qquad\square$

*Proof of Theorem 6.1.* Let $Z \sim \mathrm{Tulap}\left(T(x), b = e^{-\epsilon}, \frac{2\delta b}{1 - b + 2\delta b}\right)$. We know that the distribution of $Z$ is symmetric with location $T(x)$, and the pdf $f_Z(t)$ is increasing as a function of $|t - T(x)|$. It follows that $f_Z(t)$ has a MLR in $T(x)$. By Lemma 5.1, we know that $\phi(x) = F_Z(m)$ satisfies (2)-(5), so by Lemma 10.3, we have the desired result. $\qquad\square$

**Definition 10.4** (p-Value: CB02). For a random vector $X_i \overset{\text{iid}}{\sim} f_\theta$, a *p-value* for $H_0 : \theta \in \Theta_0$ versus $H_1 : \theta \in \Theta_1$ is a statistic $p(\underline{X})$ taking values in $[0,1]$, such that for every $\alpha \in [0,1]$,

$$\sup_{\theta \in \Theta_0} P_\theta(p(X) \leq \alpha) \leq \alpha.$$

The smaller the value of $p(X)$, the greater evidence we have for $H_1$ over $H_0$.

*Proof of Theorem 6.2.* We denote by $F_{Z \sim \theta_0}(\cdot)$ the cdf of the random variable $Z$, distributed as $Z \mid X \sim \mathrm{Tulap}(X, b, q)$ and $X \sim \mathrm{Binom}(n, \theta_0)$.

1. First we show that $p(\theta_0, Z)$ is a p-value, according to Definition 10.4. To this end, consider

$$\sup_{\substack{\theta \leq \theta_0}} P_{\substack{Z|X \sim \mathrm{Tulap}(X,b,q) \\ X \sim \mathrm{Binom}(n,\theta)}}(p(\theta, Z) \leq \alpha) = P_{\substack{Z|X \sim \mathrm{Tulap}(X,b,q) \\ X \sim \mathrm{Binom}(n,\theta_0)}}(p(\theta_0, Z) \leq \alpha)$$

   using the fact that $X$ has a monotone likelihood ratio in $\theta$. Note that $p(\theta_0, Z) = 1 - F_{Z \sim \theta_0}(Z)$. When $X \sim \mathrm{Binom}(n, \theta_0)$, we have that $p(\theta_0, Z) = 1 - F_{Z \sim \theta_0}(Z) \sim \mathrm{Unif}(0, 1)$. So,

$$P_{\substack{Z|X \sim \mathrm{Tulap}(X,b,q) \\ X \sim \mathrm{Binom}(n,\theta_0)}}(p(\theta_0, Z) \leq \alpha) = P_{U \sim \mathrm{Unif}(0,1)}(U \leq \alpha) = \alpha.$$

2. Let $N \sim \mathrm{Tulap}(0, b, q)$, and recall from Theorem 5.2 that the UMP-$\alpha$ test for $H_0 : \theta \leq \theta_0$ versus $H_1 : \theta > \theta_0$ is $\phi^*(x) = F_N(x - m)$, where $m$ satisfies $E_{\theta_0} \phi^*(x) = \alpha$. We can write $\phi^*$ as

$$\phi^*(x) = F_N(x - m) = P_{N \sim \mathrm{Tulap}(0,b,q)}(N \leq X - m \mid X)$$
$$= P_N(X + N \geq m \mid X) = P_{Z|X \sim \mathrm{Tulap}(X,b,q)}(Z \geq m \mid X)$$

   where $m$ is chosen such that

$$\alpha = E_{X \sim \theta_0} \phi^*(X) = E_{X \sim \theta_0} P_{Z|X \sim \mathrm{Tulap}(X,b,q)}(Z \geq m \mid X)$$
$$= P_{\substack{Z|X \sim \mathrm{Tulap}(X,b,q) \\ X \sim \mathrm{Binom}(n,\theta_0)}}(Z \geq m) = 1 - F_{Z \sim \theta_0}(m),$$

   where $F$ is the cdf of the marginal distribution of $Z$, where $Z|X \sim \mathrm{Tulap}(X, b, q)$ and $X \sim \mathrm{Binom}(n, \theta_0)$. From this equation, we have that $m$ is the $(1 - \alpha)$-quantile of the marginal distribution of $Z$.

Let $R|X \sim \text{Bern}(\phi^*(X))$ and $Z|X \sim \text{Tulap}(X, b, q)$. Then

$$R|X \stackrel{d}{=} I(Z \geq m) \mid X \stackrel{d}{=} I(F_{Z \sim \theta_0}(Z) \geq F_{Z \sim \theta_0}(m)) \mid X$$
$$\stackrel{d}{=} I\left(1 - \alpha \leq F_{Z \sim \theta_0}(Z)\right) | X$$
$$\stackrel{d}{=} I\left(p(\theta_0, Z) \leq \alpha\right) | X.$$

Taking the conditional expected value $\mathbb{E}(\cdot \mid X)$ of both sides gives

$$\phi^*(x) = E(R \mid X) = P_{Z|X \sim \text{Tulap}(X,b,q)}(p(\theta_0, Z) \leq \alpha \mid X).$$

3. We can express $p(\theta_0, Z)$ in the following way:

$$p(\theta_0, Z) = P_{\substack{X \sim \text{Binom}(n,\theta_0) \\ N \sim \text{Tulap}(0,b,q)}}(X + N \geq Z) = P_{X,N}(-N \leq X - Z)$$
$$= E_{X \sim \text{Binom}(n,\theta_0)} P_N(N \leq X - Z \mid X) = E_{X \sim \text{Binom}(n,\theta_0)} F_N(X - Z)$$
$$= \sum_{x=0}^{n} F_N(x - Z)\binom{n}{x}\theta_0^x(1 - \theta_0)^{n-x},$$

which is just the inner product of the vectors $\underline{F}$ and $\underline{B}$ in algorithm 1. $\qquad \square$