[Reviews · NeurIPS 2018]

Reviewer 1



Short summary: In this paper, the authors derive uniformly most powerful differentially private simple and one-sided tests for the parameter $\theta$ of the Binomial distribution with parameters $(n,\theta)$. In this context, the requirement of differential privacy translates to certain regularity conditions on the test as a function. The authors explicitly establish the necessary form of a test which fulfils these conditions (i.e. the in some sense strongest version of them) and express it as the distribution function of a novel distribution they call Tulap distribution. The UMP-property is then asserted through a Neyman-Pearson-type lemma. Later on, the close connection between the test and the Tulap and Binomial distributions leads to a convenient representation of the associated p-value and therefore the test as such. The paper closes with possible applications to situations without a priori Binomial distributions and simulations comparing the empirical type-I error probability and power when using the new test, the classical non-private UMP test and the test based on Normal approximation. Comments: Since the differentially private UMP tests for the considered problem were unknown and the authors introduce a new distribution as an essential tool, I consider the paper significant and original. Also, this work appears to be technically sound. A final proof-reading would be advisable nonetheless; for instance, there are several typos in the paragraph on the nearest integer function (lines 98-100) and there is a factor $\exp(\epsilon)$ missing in the second equation on page 11. In fact, I would like to mention as my primary concern that I found sections 2 though 4 very difficult to read - fortunately, sections 5 through 7 are well-written and helped clarify my earlier problems. More specifically, Definition 2.1. clearly creates the impression that $X$ is a parameter. Then, in Definition 2.3., $X$ ist the random object and there is a new parameter $\theta$; however, only a few lines after that, $X$ appears as the index of the test, which is very confusing since it is actually the input variable (i.e. you would typically write $\phi(X)$). After section 3, there is no motivation yet to introduce the Tulap distribution and on top of that, you are again confused about the role of $X$ since it seems to be a parameter. I understand that each consideration in these sections is an important preparation for what follows, but the current structure is unsettling, at least for me: For instance, would it not be more natural to place Lemma 5.2 at the end of section 3 and let this be the motivation for Definition 4.1? Furthermore, the notion of post-processing (where the interpretation of $X$ as a parameter is warranted) is only needed in section 7, so would it not be more natural to discuss it only in that section? I realise that many people would dismiss all this as being a question of taste and hence it does not affect my opinion that this significant and original paper should be published, but I would still appreciate a more accessible version (through a revised structure or helpful remarks). Regarding the rebuttal: I have read the other reviews and the authors' response. Even in light of the points raised by the other reviewers, I still vote for accepting the paper since the authors' response is quite convincing; I particulary appreciate how seriously the authors took my review and which adjustments they announced.

Reviewer 2



In this paper, the authors studied hypothesis testing under differentially private (DP) constraints. They showed the necessary and sufficient conditions for a hypothesis testing procedure to be DP are (2) - (5), based on which they developed tests that were DP-UMP. Their tests depended on the Tulap distribution for the randomness in DP. They also presented procedures to calculate p-values, and showed applications to distribution-free inference. Although I found the paper interesting, I also felt the presentation was not very clear. This is probably due to the strict space limit of NIPS. Within 7 pages, the authors presented more than a dozen highly technical theorems and lemmas, without going too much detail in their implications and significance. Therefore, I feel this paper is more suitable as a journal publication, such as on AOS, EJS or Bernoulli, in which the authors could go in more depth in establishing the backgrounds and the intuitions of their proposal. In addition, I suggest the authors to discuss more regarding the relationship between DP in hypothesis tests and the power of local alternatives. It seems to me that for a hypothesis testing procedure to be DP, its power in testing local alternatives could not be too high. This limits the power of the DP hypothesis testing procedures. Therefore, it is interesting to gain some intuition regarding UMP under DP setting, i.e., how does the statistical power of a DP-UMP procedure behave if the power of testing local alternatives is upper bounded.

Reviewer 3



This manuscript derived the uniformly most powerful (UMP) tests for simple and one-sided hypothesis testing on binomial data under general $(epsilon, delta)$ privacy constraints. Specifically, the authors proposed a new probability distribution called Truncated-Uniform-Laplace (Tulap) and showed that the UMP test can be reformulated as post-processing the Tulap random variable. Based on this idea, the authors can also obtain the exact p-values and verify the superiority via simulations. I like this manuscript due to the following reasons: 1. The exact UMP test is obtained for binomial data under general $(epsilon, delta)$ privacy constraints for any parameters $\epsilon, \delta$; 2. The authors proposed the Tulap distribution, which is a generalization of the staircase and discrete Laplace distributions and could be of independent interest in the field of differential privacy. However, there are also some weaknesses for this manuscript: 1. The resulting UMP test under privacy constraint is quite straightforward and did not really reflect new ideas. Consider the traditional case without privacy constraint; by Neyman--Pearson we know that $\phi_x = 0$ for $x$ below some threshold, and then $\phi_x$ jumps to one. Now adding the privacy constraint, we cannot make $\phi_x$ increase too sharply. Then the natural idea is to just have $\phi_x$ increase as quickly as possible, subject to the privacy constraints Eqn. (2) - (5), which turns out to be optimal and also easy to prove in theory. Hence, the resulting UMP is not surprising to me in terms of both the idea and the underlying theory. 2. The UMP test construction is quite ad-hoc for the binomial model. Could this idea be generalized to other statistical models with privacy constraint as well? Overall speaking, I still like this manuscript mainly due to the introduction of the Tulap distribution and its potential usage in future works, and I would appreciate the authors' rebuttal on the weaknesses. Edit: I have read the authors' rebuttal and feel they are quite satisfactory to me. I choose to raise my score from 6 to 7.